# Essential Oils Distilled from Colombian Aromatic Plants and Their Constituents as Penetration Enhancers for Transdermal Drug Delivery

**DOI:** 10.3390/molecules28062872

**Published:** 2023-03-22

**Authors:** Heider Carreño, Elena E. Stashenko, Patricia Escobar

**Affiliations:** 1Departamento de Ciencias Básicas, Centro de Investigación en Enfermedades Tropicales (CINTROP), Escuela de Medicina, Universidad Industrial de Santander, Piedecuesta 681011, Colombia; 2Centro de Investigación en Biomoléculas (CIBIMOL), Escuela de Química, Universidad Industrial de Santander, Bucaramanga 680002, Colombia

**Keywords:** permeation enhancers, essential oils, *Lippia origanoides*, *Turnera diffusa*, plant secondary metabolites

## Abstract

The study aimed to determine the enhanced effects of essential oils (EOs) and plant-derived molecules (PDMs) as penetration enhancers (PEs) for transdermal drug delivery (TDD) of caffeine. A 1% *w*/*w* solution of eight EOs and seven PDMs was included in the 1% caffeine carbopol hydrogel. Franz diffusion cell experiments were performed using mice with full-thickness skin. At various times over 24 h, 300 μL of the receptor were withdrawn and replaced with fresh medium. Caffeine was analyzed spectrophotometrically at 272 nm. The skin irritation effects of the hydrogels applied once a day for 21 days were investigated in mice. The steady-state flux (J*_SS_*) of the caffeine hydrogel was 30 ± 19.6 µg cm^−2^ h^−1^. An increase in caffeine J*_SS_* was induced by *Lippia origanoides* > *Turnera diffusa* > eugenol > carvacrol > limonene, with values of 150 ± 14.1, 130 ± 47.6, 101 ± 21.7, 90 ± 18.4, and 86 ± 21.0 µg cm^−2^ h^−1^, respectively. The Kp of caffeine was 2.8 ± 0.26 cm h^−1^, almost 2–4 times lower than that induced by *Lippia origanoides* > *Turnera diffusa* > limonene > eugenol > carvacrol, with Kp values of 11 ± 1.7, 8.8 ± 4.2, 6.8 ± 1.7, 6.3 ± 1.2, and 5.15 ± 1.0 cm h^−1^, respectively. No irritating effects were observed. *Lippia origanoides*, *Turnera diffusa*, eugenol, carvacrol, and limonene improved caffeine’s skin permeation. These compounds may be as effective as the PE in TDD systems.

## 1. Introduction

Overcoming the skin barrier (or certain layers of the skin) for drug entry without deleterious skin reactions (i.e., skin irritation or an allergic reaction) is still a major challenge to resolve. The skin is a complex structure formed by epidermal, dermal, and hypodermal layers that acts as a remarkably effective barrier to the entrance of exogenous agents, including drugs. Its barrier function is due to the stratum corneum (SC), the most superficial layer of the skin, described as a “brick-and-mortar” structure consisting of tightly arranged corneocytes enveloped by an intercellular lipid matrix [1]. Transdermal drug delivery (TDD) systems aim to change or destabilize the SC safely, allowing drug access through transepidermal (inter/transcellular) or appendageal (hair follicles, sebaceous glands, and sweat ducts) SC-allowed pathways [2]. Several TDD systems such as drug-loaded patches, liposomes, transferosomes, solid lipid nanoparticles, micelles, polymeric nanoparticles, emulsions, or physical methods such as microneedles, thermal ablation, ultrasounds, or iontophoresis have been studied for their ability to improve drug skin permeation and deposition [2,3]. In addition, the use of chemical penetration enhancers (CPEs) is used in topically applied formulations and constitutes one of the more simplistic and ample strategies for TDD [4]. They are substances that rapidly and reversibly promote percutaneous penetration of drugs by interacting with SC intracellular lipids or proteins (lipid-protein-partitioning theory) or by increasing the solubility of the drug in SC lipids. They modified the solubility characteristics of these lipids and facilitated drug transport, partitioning, and diffusion toward deeper layers of the skin [4]. A wide range of molecules, including glycerides, azone, urea, pyrrolidones, sulfoxides, fatty acids, and some essential oils (EOs) and their constituents, have been used as CPEs for TDD [4,5,6].

The focus has been on identifying effective penetration enhancers (PE) from natural sources. For example, essential oils and terpenes are one of the encouraging groups of candidates to be used as clinically acceptable PE [4,5]. Essential oils are natural products distilled from aromatic plants and composed of a complex mixture of small, hydrophobic, and volatile molecules, where terpenoids are usually their major components. EOs and their volatile constituents, especially mono- and sesquiterpenoids, alone or in mixtures, have been extensively studied in PEs [4,5]. They have been included in different types of topical formulations, free or encapsulated, in diverse nano TDD systems, such as some of those that were previously named [7]. Terpene-rich EOs such as eucalyptus oil (1,8-cineole), clove oil (eugenol), peppermint oil (menthol and menthone), limonene, turpentine oil (α-pinene and β-pinene), and menthol have been identified as the most frequently used compounds for PEs [8], however, others such as thyme, basil, angelica, and fennel EOs have also demonstrated interesting PE activities [5,9]. One of the most common methodologies for studying TDD in vitro or ex vivo is the diffusion cell system (Franz cells) [10]. They consist of a donor and a receptor compartment, separated by a barrier, i.e., an artificial membrane or skin sample. Diffusion studies testing EOs or their constituents as skin PEs for TDD of several drugs have been performed using different types of membranes, such as excised human, pig, rabbit, rat, mouse, and hairless mouse skin, and cellulose or artificial membranes [5]. Ranked by their drug-lipophilicity (LogP), EOs have been successful in delivering a range of drugs across different types of skin, including aminophylline (MW 420.4, LogP −3.03), 5-fluorouracil (MW 130.1, LogP −0.89), metoprolol succinate (MW 267.36, LogP 2.2), chlorhexidine digluconate (MW 897.8, LogP 2.7), trazodone hydrochloride (MW 408.3, LogP 3.1), ibuprofen (MW 206.2, LogP 3.97), ketoconazole (MW 531.4, LogP 4.3), and diclofenac sodium (MW 318.1, LogP 4.5) [5]. They can enhance drug penetration into the lower skin layers by interacting with lipids and proteins, i.e., by disintegrating (solubilizing and extracting) the intercellular lipid structure between corneocytes in SC, by interacting with an intercellular domain of protein, i.e., keratin, or by increasing the partitioning of a drug, a coenhancer, water, or any combination of these [5,6,11]. Overall, they could be safe (free of adverse side effects); however, toxicity is dependent on many factors such as their chemical composition, purity (method of extraction and conservation, age), and dosage, where a balance of safety versus potency allows their use at the lowest possible concentration [5,6,12].

As part of a research program based on the discovery of multifunctional biological activities of Colombian essential oils and their constituents, in this case, those related to the improvement of some skin conditions that occur within the deeper layers of the skin and internal organs, this study aimed to determine the enhancing effect of EOs and PDMs obtained from Colombian plants as skin PEs of caffeine [13], a hydrophilic compound suitable for TDD (MW 194.2, LogP = −0.07). Two EOs and three PDMs were safely included in a carbopol hydrogel enhancer by 2–4 caffeine permeations.

## 2. Results and Discussion

### 2.1. Essential Oil Characterization

The essential oils studied here were obtained by microwave-assisted hydrodistillation from eight aromatic plants cultivated at the Agroindustrial Pilot Complex and Botanical Garden (“Un mundo en un jardín” in Spanish, “A world in a garden”) at CENIVAM (the National Centre for Research in Agroindustrialization of Tropical Medicinal Aromatic Plants at the Industrial University of Santander, Bucaramanga, Colombia). The EO yields varied from 0.1 to 1.2% (Table 1). The highest EO yields were obtained for *Lippia origanoides* (Verbenaceae) plants (0.5–1.2%). EO chemical characterization was performed by GC/MS with electron ionization (EI, 70 eV) and both polar and nonpolar capillary columns, using many standard compounds as well for confirmatory identification. In Appendix A, the identification, linear retention indices (LRIs), and relative amounts (>1%) of the principal EO constituents are reported. Appendix A shows *L. origanoides* (Verbenaceae) and *Turnera diffusa* (Passifloraceae) plants, and Appendix A displays eight chromatographic profiles of the EOs studied. The main EO constituents were monoterpenoids, sesquiterpenoids, and phenolic monoterpenes (thymol, carvacrol, and their methyl ethers and acetates). Among common compounds found in most EOs studied, α- and β-pinenes, α- and β-phellandrenes, limonene, 1,8-cineole, p-cymene, β-myrcene, γ-terpinene, trans-β-caryophyllene, and its oxide were present (Table 1). Three *L. origanoides* chemotypes were differentiated according to their main compounds, as follows: (1) *Phellandrene*-chemotype (EO3, EO9); in these EOs, α- and β-phellandrenes with trans-caryophyllene and its oxide prevailed; (2) *Carvacrol*-chemotype (EO8); and (3) *Thymol*-chemotype (EO19); in the latter EOs, the amount of phenolic compounds (carvacrol, thymol, and their ethers and esters) represented more than 50–70% of the total composition. Sesquiterpene hydrocarbons were the main constituents of the *Turnera diffusa* EO, while monoterpene hydrocarbons and their oxygenated derivatives dominated the EOs distilled from *Steiractinia aspera* (EO1), *Calycolpus moritzianus* (EO4), *Piper anduncum* (EO5), and phellandrene-chemotype *L. origanoides* EOs (EO3, EO19).

### 2.2. Gel Characterization and Stability

Hydrogels of caffeine alone and with EO or PDM were easily prepared (Table 2). All formulations were homogeneous and slightly whiter after EO or PDM addition. The viscosity of caffeine gels plus EO or PDM was statistically higher (*p* < 0.05) than that of gels with caffeine alone (values of 251.234 and 303.935 cP and 179.587 ± 2904.8 cP, respectively). The average pH was 5.5 and stable for up to 30 days, except for caffeine plus PDM22, which slightly decreased from 5.5 to 4.5 after 7 days. Since natural skin surface pH is acidic on average (pH of 4.7, i.e., below 5) [14], prepared gels were suitable for topical application on ecdermic skin and potentially with no or low risk of toxicological effects. No changes in caffeine concentration values between samples (12.73 ± 2.13 µg/mL) and controls (average 11.6 ± 0.20 µg/mL) at 1, 15, and 30 days after preparation were found, indicating uniformity in the drug content within and between the formulations (Table 2). The stability of pH values between samples and controls could favor both caffeine maintenance in its nonionized form and homogeneous conditions in the case of a pH-sensitive (fast or delayed) caffeine release [15,16]. The use of carbopol 940^®^ as a gelling agent was an important achievement; it is a crosslinked polyacrylic polymer that is acidic but neutralized with an appropriate base (i.e., triethanolamine) to achieve its thickening property. Carbopol-hydrogels are three-dimensional polymer networks that are biocompatible, adhesive, and biodegradable and can absorb large amounts of water and load large amounts of water-soluble drugs. In addition, similar to smart hydrogels, they can control drug release by responding to stimuli such as pH, temperature, and light [15,16]. Since EOs and their constituents are volatile and easily decomposed by environmental conditions, novel TDD nanosystems such as microemulsions, nanoemulsions, nanoemulgels, liposomes, solid lipid nanoparticles, and nanostructured lipid carriers have been designed to increase their effectiveness as skin PE. As an example, both lipophilic and hydrophilic drugs such as caffeine (MW194.2, LogP −0.07), naproxen (MW 230.3, LogP 3.2), and diclofenac sodium (MW 318.2, LogP 4.8) markedly enhanced skin permeation when using nanoemulsion or nanoemulgel formulations containing eucalyptol, oleic acid, and cumin EO as PE, in comparison to all controls [17,18].

### 2.3. Caffeine Model

The cumulative amount (µg cm^−2^) of caffeine that permeates across the skin throughout the study (0–24 h) without EO or PDM had an average of 619.35, SD = 236,56 µg cm^−2^ (from three technical replicates), with a steady-state flux (J*_SS_*) of 32.28 ± 3.25 µg cm^−2^ h^−1^ (*n* = 12) (Figure 1 and Figure 2). Caffeine was selected as a model compound with no intention of transdermal/topical therapeutic application. Its use as a model has been recommended by the Organization for Economic Cooperation and Development (OECD) guideline 428 [13].

### 2.4. EO as a Permeation Enhancer

The effect of 1% *w*/*w* EO on the cumulative amount of caffeine that permeates across the skin and the resulting skin permeation parameters are shown in Figure 1 and Figure 2 and Table 3. Among the EOs, EO19 (*Lippia origanoides*, Thymol chemotype) and EO2 (*Turnera diffusa*) significantly enhanced the J*ss* values and permeability coefficients (Kp) with an enhancement index (EI) of 3.7 and 3.0, respectively. In contrast, the lowest enhancement in caffeine flux was obtained with EO1 (*Steiractinia aspera*), EO4 (*Calycolpus moritzianus*), and EO5 (*Piper anduncum*) with EI values lower than 0.9. *Lippia origanoides* H.B.K. (Verbenaceae) is an aromatic plant growing in different Colombian regions and used in traditional medicine due to its antioxidant, anti-viral, anti-genotoxic, anti-parasitic, and repellent activities [19,20]. As revealed by chemical analysis, two of their different chemotypes [21] were tested in this work. The phellandrene chemotypes EO3 and EO9 were the least caffeine PE compounds, with EI values of 1.15 and 1.41, respectively, and the carvacrol (EO8) and thymol (EO19) chemotypes were the best, with EI values of 1.6 and 3.7, respectively. The higher enhancement in caffeine permeation induced by EO19 could be due to its monoterpene constituents ranking from thymol > *p*-cymene > carvacrol > β-myrcene and γ-terpinene (Table 1).

The other promising EO was distilled from the *Turnera diffusa* (damiana) (Passifloraceae) plant. It is a medicinal plant used traditionally as an aphrodisiac, especially in Central America, and is effective against diabetes and gastric, skin, and respiratory diseases [22]. In contrast with EO19, its main EO constituents were mostly bicyclic sesquiterpenes (except β-elemene), in descending order: dehydrofukinone > epi-aristolochene > β-selinene > valencene > β-elemene > trans-β-caryophyllene (Table 1). Only the last one was evaluated as a PE in this study (PMD41) and displayed an EI value of 1.23. We did not find information about *T. diffusa*-EO as a PE; however, some PE activities of valencene and other sesquiterpenes such as farnesol and neridol have been described for drugs such as 5-fluorouracil (MW 130.1, LogP −0.9), propranolol hydrochloride (MW 295.8, LogP −0.5), and valsartan (MW 435.5, LogP 4.5) [5,6,23]. To date, the Colombian *T. diffusa* EO main constituents have been related to interesting activities such as anesthetic (dehydrofukinone) and anti-cancerogenic (β-elemene) activities and acting as a precursor of phytoalexins such as capsidiol (5-epi-aristolochene) [24,25,26].

### 2.5. PDM as a Permeation Enhancer for Caffeine

The results of the enhancement of caffeine permeation by 1% *w*/*w* PMD are shown in Figure 1 and Figure 2 and Table 3. Among the PDMs, PMD10 > PDM22 > PDM6 > PDM19 (bromonaphthalene > eugenol/carvacrol/limonene) with EI 2.97, 2.6, 2.4, 1.96 but not PMD4, PMD35, or PMD41 (citronellal/α-phellandrene/trans-β-caryophyllene) led to a statistically significant (*p* < 0.05) increase in caffeine flux compared with caffeine gel alone. No significant difference was found between the mean values of lag time. Penetration from caffeine gels containing PMD was greater than that of the control, displaying an EI higher than 1 (Table 3). PMD10 was the best PE compound. It was included in a list of the initial 40 PMDs sent to our lab for phenotypic screening assays for wound healing or anti-leishmanial compound candidates. PMD10 is a naphthalene organic compound composed of a fused pair of benzene rings and bromide (Table 4). Some polyaromatic hydrocarbons such as naphthalene have been found in floral extracts of *Magnolia* species and the calli of *Origanum vulgare* spp. Virens [27,28]. However, they are highly toxic to pulmonary tissues [29]. PMD10 was not included in our conclusion because of its toxicity.

Except for PMD10, our results corroborate the role of a variety of terpenes as PE [5,6]. Eugenol was the most effective (EI of 2.97). It is the main component in clove oil and has been amply used in pharmaceuticals, dentistry, food, cosmetics, and as a local antiseptic and analgesic due to its recognized biological properties such as anti-inflammatory, antioxidant, anti-cancer, anti-fungal, and anti-bacterial activities [30]. Eugenol and clove oil (81% eugenol) are some of the most studied compounds known as PE [8]. They enhance the skin permeation of drugs such as 1–3% valsartan (MW 435.5, LogP 4.5) and ibuprofen (MW 206.3, LogP 4.0) by 2–6 times [9,31,32].

Carvacrol was also an important PE molecule demonstrated in this study (two- to three-fold enhancement of caffeine permeation). Some carvacrol formulations containing 10% carvacrol in propylene glycol or 3 mM carvacrol in PBS enhanced testosterone (MW 288.4, LogP 3.3) and corticosterone (MW 346.5, LogP 1.9) permeate through human skin by 7 to 10 times. In addition, a high enhancement (approximately 38 times) of azidothymidine (MW 267.2, LogP 0.05) and increased flow and coefficient of permeability for haloperidol (MW 375.9, LogP 4.3) in formulations containing 5% carvacrol/alcohol have been reported [33,34,35]. Carvacrol is a phenolic monoterpene found in EOs from some *Lippia* (Table 1), *Origanum*, and *Thymus* species with interesting pharmacological effects such as analgesic, anti-inflammatory, anti-tumor, and antioxidant activities [36]. No available data on carvacrol-rich *Lippia* EO as PE or TDD was found.

Finally, limonene was also an important caffeine PE. Limonene is one of the most studied drug-enhancer terpenes [5]. A dose-response effect of limonene as a PE has been demonstrated with compounds such as bufalin (MW 386.5, LogP 2.8), hydrocortisone (MW 362.5, LogP 1.6), haloperidol (MW 375.9, LogP 4.3), ketoprofen (MW 254.3, LogP 3.1), and genistein (MW 270.2, LogP 2.9), displaying a decreasing EI of 28.1, 28.0, 26.5, 2.0, and 1.7, respectively [5,6,37,38].

### 2.6. Skin Irritation Assessment

In this work, mice showed no irritation signs or skin edema after treatment. Carbopol hydrogels containing caffeine alone or caffeine plus EO19, EO2, PDM6, PMD19, and PMD22 at 1% were safe. The treated skin was intact; no inflammation or erythema compared to the untreated site was observed (score 0 in all mice and EO at all evaluated times). In a previous study, using different *L. origanoides* chemotypes, we demonstrated the skin irritancy of 100–20% thymol- and carvacrol-chemotypes (but not phellandrene) used topically but used at 10% or less, and any change both macro and at the histopathological level was detected [39]. The use of eugenol, carvacrol, and limonene terpenes as flavoring agents or adjuvants has been recognized as safe (GRAS) by the United States Food and Drug Administration [40]. However, the overall use of EOs or PMDs as PE in TDD systems must be executed with precaution due to their potential toxicity at higher concentrations and the possibility of acute (i.e., contact dermatitis and allergic reactions) or long-term effects.

## 3. Materials and Methods

### 3.1. Plant Material, Essential Oil Distillation, and Analysis

Eight EOs were distilled at the CENIVAM Experimental Agro-Industrial pilot complex, Universidad Industrial de Santander, Colombia, from *Steiractinea aspera* (EO1), *Turnera diffusa* (EO2), phellandrene- (EO3, EO9), carvacrol- (EO8), and thymol- (EO19) *Lippia origanoides* chemotypes, *Calycolpus moritzianus* (EO4), and *Piper anduncum* (EO5) aromatic Colombian plants. The ex situ plant collection was kept under controlled agricultural conditions (26–28 °C, 75–80% humidity), and only healthy and undamaged plants, previously chopped, were used immediately for their distillation. A contract for access to genetic resources and derived products for bioprospecting purposes, No. 270, was signed between the Ministry of Environment and Sustainable Development and Universidad Industrial de Santander.

Essential oil extraction was performed by microwave-assisted hydrodistillation using Clevenger equipment adapted to a microwave system as described elsewhere [18]. The major volatile secondary metabolites present in each EO were identified and quantified by gas chromatography coupled to mass spectrometry (GC/MS) as described by Stashenko and Martinez [16]. The codes, EO yields, relative amounts (%), and composition are described in Table 1. The identification of the main EO constituents was performed by comparison of their tR, LRIs (Appendix A), and mass spectra (molecular ions, fragmentation patterns, isotopic ratios) with those of the standard compounds analyzed under the same GC/MS experimental conditions. For the GC/MS analysis, two capillary columns of the same dimensions (L: 60 m, I.D.: 0.25 mm, df: 0.25 µm) but of different polarities (nonpolar column, DB-5MS, with a 5%-Ph-PDMS, and a polar column, DB-WAX, with a PEG, stationary phases) were used. Both columns were acquired from J&W Scientific (Folsom, CA, USA). An Agilent Technologies (AT) gas chromatograph, the GC 6890 Plus (AT, Palo Alto, CA, USA), coupled to a mass selective detection system, the MSD 5973 Network (AT, Palo Alto, CA, USA), was employed. The EOs were injected at 250 °C in a split mode (30:1) using helium (99.995%) as the carrier gas at 1 mL min^−1^ constant flow. The GC oven temperature was programmed as follows: from 45 °C (5 min) to 150 °C (2 min) at 4 °C/min, then to 300 °C (10 min) at 5 °C/min for the nonpolar DB-5MS column, and from 50 °C (5 min) to 150 °C (7 min) at 4 °C/min, then to 230 °C (50 min) at 4 °C/min) for the polar DB-WAX column. All experimental data (GC peak integration, mass spectra, library search) were processed using MSD ChemStation G1701DA software (AT, Palo Alto, CA, USA).

### 3.2. Chemicals

Caffeine, citronellal, limonene, bromonaphthalene, carvacrol, eugenol, α-phellandrene, trans-β-caryophyllene, carbopol 940, sodium benzoate, and triethanolamine were purchased from Sigma-Aldrich (Milwaukee, WI, USA). All the reagents were of analytical grade. The code, structure, molecular weight (MW), and LogP of the PDMs are shown in Table 4.

### 3.3. Preparation of Gels and Caffeine Quantification

Hydrogels were prepared with constant stirring by dispersing 1.0% (*w*/*w*) of Carbopol 934 in distilled, deionized water. Caffeine (1.0% *w*/*w*), sodium benzoate (0.1%), and EO or PMD at 1.0% *w*/*w* were added. The stirring was continued for 30 min and thereafter neutralized using 0.5% triethanolamine until a transparent gel appeared. The gels were inspected visually for their color, homogeneity, consistency, and phase separation. The pH was measured using a digital pH meter, and the viscosity was measured using a viscometer (Atago, Japan). The final concentration of caffeine was analyzed by UV-Vis spectrophotometry at 272 nm. For a calibration curve, a standard stock solution of caffeine was prepared by dissolving 100 mg of caffeine in 100 mL of distilled water and diluting 1:10 with distilled water (100 µg/mL). Standard solutions of 1–50 µg/mL of caffeine in distilled water were read at 272 nm, and the absorbance calibration curve was plotted (concentration versus absorbance). The correlation coefficient was 0.99 (y = 0.0425x + 0.1343).

### 3.4. Mice

Mice were supplied by the Colombian National Health Institute. They were housed with a 12 h light/dark cycle at 23 °C and 55 ± 5% relative humidity, with access to water and mouse food pellets ad libitum. All procedures were performed under a protocol approved by the Ethics Committee of the Industrial University of Santander, Bucaramanga, Colombia (CIENCI, Code 4110).

### 3.5. Permeation Studies

#### 3.5.1. Skin Membrane

BALB/c mice were trichotomized, and after 24 h, they were anesthetized by intraperitoneal injection of a ketamine/xylazine cocktail and euthanized by cervical dislocation. Dorsal full-thickness skin was obtained, and the subcutaneous tissue was carefully removed and washed in phosphate-buffered saline (PBS) with a pH of 7.4 twice. Skin samples were wrapped in aluminum foil and stored in a freezer at −20 °C until use, not exceeding 3 months. On the day of the experiment, the skin was left at room temperature and cut into pieces, and skin samples were mounted over the Franz cells with the SC side facing the donor compartment.

#### 3.5.2. Transdermal Delivery of Caffeine

Franz diffusion cells (PermeGear Inc., Hellertown, PA, USA) with a diffusional effective area of 0.196 cm^2^ and a fluid receptor of PBS pH 7.4 were used. The receptor compartment of 3 mL PBS at pH 7.4 was continuously mixed using a magnetic stirring bar (300 rpm) at 32 °C. Three hundred milligrams of caffeine alone, caffeine plus EO, or PMD gels (equivalent to 3 mg of caffeine) were placed on the membrane surface in the donor compartment and sealed with parafilm to avoid evaporation. Samples of 300 µL were withdrawn at 0, 1, 2, 4, 6, and 24 h from the receptor phase and immediately replaced with the same volume of the receptor fluid. The amount of caffeine was analyzed spectrophotometrically at 270 nm by the validated UV-Vis method described previously. The integrity of the mouse membrane was examined after the experiment. Skin membranes were fixed using 10% buffered (phosphate buffer) formalin and embedded in paraffin blocks. Sections (5 µm in thickness) were stained with hematoxylin-eosin (H–E) and examined microscopically.

#### 3.5.3. Experimental Data

The data obtained were analyzed for a cumulative amount of drug permeated (µg cm^−2^), steady-state flux (J*ss,* µg cm^−2^ h^−1^), permeability coefficient (cm h^−1^), and lag time (h). The cumulative amount (Q, μg cm^−2^) of caffeine permeated across the skin (corrected for acceptor phase replacement) was plotted against time (h), and the steady-state flux J*_SS_* (μg cm^−2^ h^−1^) was calculated from the linear region of the slope of the plot. The permeability coefficients (Kp) were calculated according to the equation: Kp = J*ss*/Cd, where Cd is the initial drug concentration. The lag time was interpreted as the intercept with the *x*-axis. The influence of PDM and EO on the caffeine permeation profile (penetration-enhancing activities) was expressed as the enhancement index (EI), i.e., the ratio of the Kp value with an enhancer to that obtained without an enhancer.

### 3.6. Skin Adverse Effect Determination

To examine whether the 1% EO and PDM hydrogels induced skin irritation/corrosion, a test was performed using BALB/c mice (n = 1). Approximately 24 h before the test, fur was shaved from the dorsal area, and caffeine and caffeine plus 1% EO or PDM hydrogels were gently applied over the shaved area. A separate, untreated site was used as a control. The treatment was performed once a day for 21 days. Signs of edema or erythema at the application site after 4, 24, and 72 h and after 9 and 14 days post-treatment were registered and scored from 0 (no irritation) to 4 (severe irritation).

### 3.7. Statistical Analysis

Using the statistical program GraphPad Prism 8.1 (GraphPad Software Inc., San Diego, CA, USA), one-way analysis of variance (ANOVA) was used to analyze each experiment, along with post hoc comparisons (Tukey Dunnett’s test). Comparisons were made between the caffeine gel formulations without and with EO or PMD. A result was considered significant when *p* was less than 0.05. Data are presented as the means ± SEMS, and the number of replicates (n) is given in the pertinent figures.

## 4. Conclusions

The findings demonstrate that thymol-rich *L. origanoides* (EO19) followed by sesquiterpene-rich *T. diffusa* (EO2) enhanced caffeine delivery via the transdermal route, still higher than eugenol, carvacrol, and limonene, which are recognized and confirmed PE terpenoids. No skin irritation was observed when 1% EOs and PDM hydrogels were used for at least 15 days. Many publications revised in this paper provide substantial evidence that some EO and their terpene constituents are interesting PEs for TDD of low-MW (MW < 500, except for ketoconazole and chlorhexidine digluconate) and hydrophilic or lipophilic (LogP from −3.03 to 4.5) drugs. However, important contributions of this work lie in the fact that our enhancer formulations were still effective at low EO or terpene concentrations, were non-irritants, and were potentially able to be used in humans in the case of prolonged treatments (21 days). In addition, *L. origanoides* and *T. diffusa* plants are now cultivated in some Colombian regions to produce EOs at industrial scale for their analysis, applications, and commercial use.

## Figures and Tables

**Figure 1 molecules-28-02872-f001:**
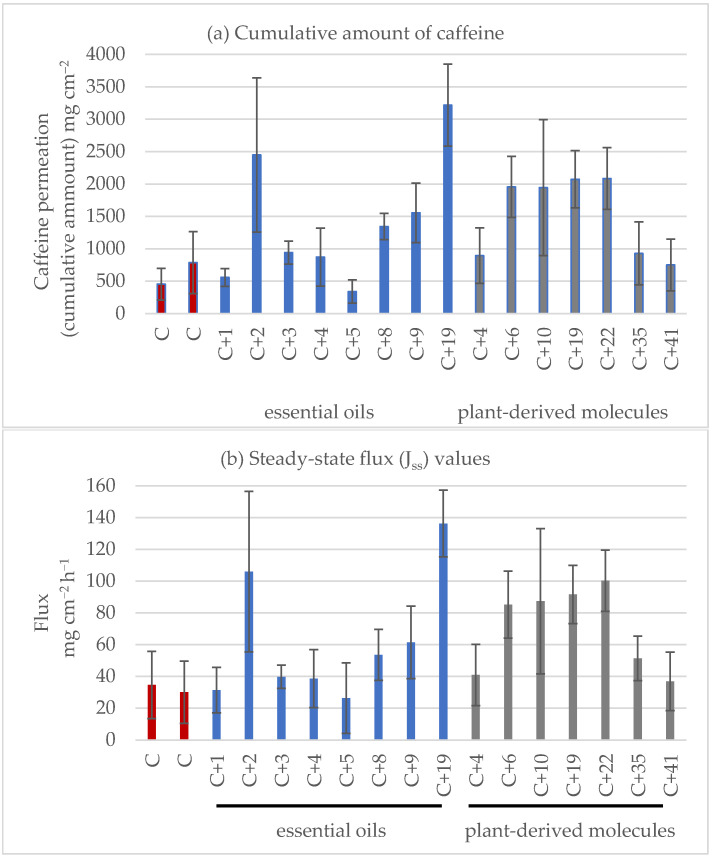
Caffeine (C) permeation through full-thickness mouse skin on caffeine gels at 24 h containing different essential oils (EOs) and plant-derived metabolites (PDMs). (**a**) The cumulative amount of caffeine and (**b**) steady-state flux (J*_ss_*) values. The results obtained with the 1% *w*/*w* caffeine gel formulation (control without permeant) are shown in red bars; formulations of 1% *w*/*w* caffeine plus 1% *w*/*w* EO 1,2,3,4,5,8,9, and 19 are shown in blue bars; formulations of 1% *w*/*w* caffeine gel plus plant-derived metabolites 4, 6, 10, 19, 22, 35, and 41 are shown in grey bars. Data are shown as the means ± SEMs, *n* = 6.

**Figure 2 molecules-28-02872-f002:**
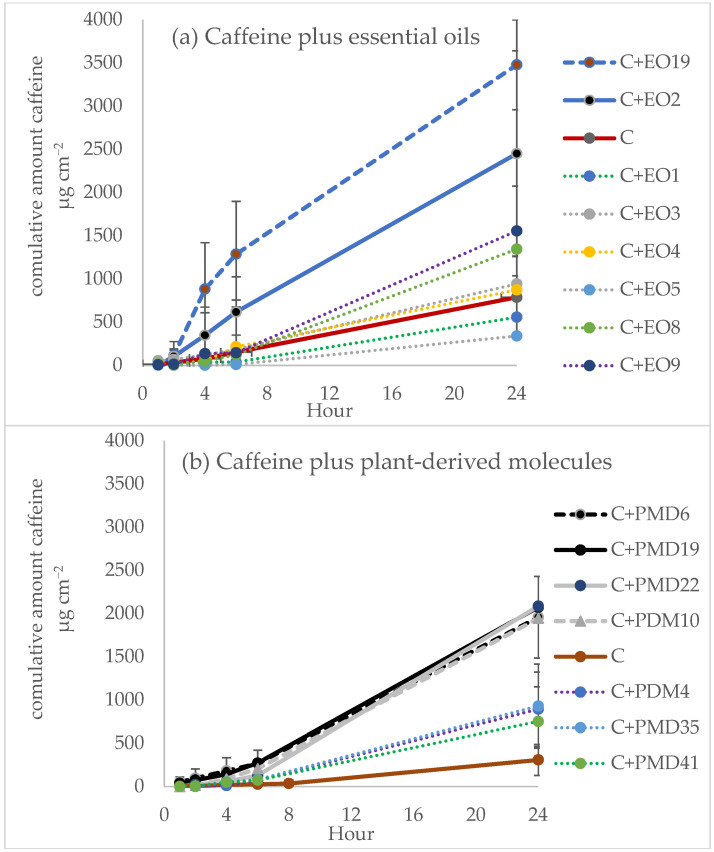
Skin permeation profiles of caffeine gel through full-thickness mouse skin at different times. Formulations of caffeine plus (**a**) essential oils (EOs) and (**b**) plant-derived metabolites (PDMs) Mean ± SD (*n* = 6). The codes for each permeant are shown on the table.

**Table 1 molecules-28-02872-t001:** Relative amounts (%) of the main compounds in the essential oils studied.

Code	Plant	Main Volatile Secondary Metabolites (>2%)
EO1	*Steiractinia aspera* Cuatrec	α-Pinene (24.9), sabinene (4.6), β-pinene (14.8), α-phellandrene (6.3), p-cymene (4.5), limonene (2.4), β-phellandrene (10.1), α-copaene (2.6), trans-β-caryophyllene (3.1), germacrene D (13.1%). EO yield—0.1%.
EO2	*Turnera diffusa*, Willd. ex Schult	p-Cymene (3.0), β-elemene (4.0), trans-β-caryophyllene (4.0), aristolochene (17.9%), β-selinene (5.2), premnaspirodiene (3.7), valencene (7.4), α-selinene (2.4), caryophyllene oxide (3.2), guaiol (3.5%), germacra-4,5,10-trien-1α-ol (3.5), dehydrofukinone (25.4). EO yield—0.3%.
EO3	*Lippia origanoides* H.B.K. Phellandrene chemotype	α-Pinene (2.0), camphene (2.5), α-phellandrene (9.3), p-cymene (8.7), limonene (4.4), β-phellandrene (3.1), 1,8-cineole (6.5), trans-β-caryophyllene (18.6), α-humulene (10.2), germacrene D (2.2), δ-cadinene (2.0), caryophyllene oxide (3.8). EO yield—0.5%.
EO4	*Calycolpus moritzianus* (O.Berg) Burret	α-Pinene (5.1), limonene (17.6), 1,8-cineole (19.1), linalool (1.3), α-copaene (3.2), trans-β-caryophyllene (6.3), viridiflorene (2.7), selina-3,7 (11)-diene (2.8), trans-nerolidol (3.5), viridiflorol (5.7), trans-geranyl-linalool (4.0). EO yield—0.2%.
EO5	*Piper aduncum* Linnaeus	α-Pinene (4.6), α-phellandrene (4.4), p-cymene (3.0), limonene (6.0), 1,8-cineole (3.6), piperitone (14.8), α-copaene (2.9), trans-β-caryophyllene (7.4), 9-epi-trans-β-caryophyllene (1.1), δ-cadinene (5.5), caryophyllene oxide (3.8), viridiflorol (5.8). EO yield—0.2%.
EO8	*L. origanoides* H.B.K. Carvacrol chemotype	β-Myrcene (2.5), p-cymene (14.4), 1,8-cineole (1.3), γ-terpinene (5.4), thymol (7.8), carvacrol (36.0), carvacryl acetate (2.0), trans-β-caryophyllene (4.4). EO yield—1.1%.
EO9	*L. origanoides* H.B.K. Phellandrene chemotype	α-Phellandrene (7.1), p-cymene (12.6), limonene (2.1), 1,8-cineole (13.0), γ-terpinene (2.4), thymol (14.0), trans-β-caryophyllene (15.1), α-humulene (8.1), caryophyllene oxide (2.5), β-eudesmol (2.6). EO yield—0.6%.
EO19	*L. origanoides* H.B.K. Thymol chemotype	β-Myrcene (2.1), p-cymene (10.7), γ-terpinene (2.0), thymyl methyl ether (1.0), thymol (72.3), carvacrol (4.4), EO yield—1.2%.

**Table 2 molecules-28-02872-t002:** Caffeine gels contain essential oils (EOs) and plant-derived metabolites (PDMs).

Gel	Color/Appearance/ Homogeneity	Caffeine Concentration (µg/mL)	pH		
			Days after Preparation		
			1	7	15
G-Vehicle-1	Neutral/translucent	11.46	5.5	5.5	5.5
G-Vehicle-2	Neutral/translucent	11.74	5.5	5.5	5.5
G-EO-1	White/opaque	11.90	5.5	5.5	5.5
G-EO-2	White/opaque	12.92	5.5	5.5	5.5
G-EO-3	White/opaque	11.68	5.5	5.5	5.5
G-EO-4	White/opaque	13.92	5.5	5.5	5.5
G-EO-5	White/opaque	9.71	5.5	5.5	5.5
G-EO-8	White/opaque	12.59	5.5	5.5	5.5
G-EO-9	White/opaque	14.63	5.5	5.5	5.5
G-EO-19	White/opaque	12.26	5.5	5.5	5.5
G-PDM-4	White/opaque	11.35	5.5	5.5	5.5
G-PDM-6	White/opaque	11.31	5.5	5.5	5.5
G-PDM-10	Neutral/translucent	11.26	5.5	5.5	5.5
G-PDM-19	White/opaque	17.79	5.5	5.5	5.5
G-PDM-22	White/opaque	16.12	5.5	4.5	4.5
G-PDM-35	White/opaque	14.01	5.5	5.5	5.5
G-PDM-41	White/opaque	11.41	5.5	5.5	5.5

**Table 3 molecules-28-02872-t003:** Effect of essential oils (EOs) and plant-derived metabolites (PDMs) at 1% *w*/*w* on the permeation parameters of caffeine using mouse skin (*n* = 6).

Gel	(Mean ± SD)			EI
	Flux (J*_ss_*) (µg cm^−2^ h^−1^)	Kp (cm h^−1^)	Tlag (h)	
G-vehicle-1	34.57 ± 21.19	2.98 ± 1.83	2.08 ± 1.13	1.00
G-Vehicle-2	29.98 ± 19.64	2.61 ± 1.71	2.55 ± 1.09	1.00
G-EO-1	31.27 ± 14.30	2.58 ± 1.18	1.22 ± 1.71	0.87
G-EO-2	105.92 ± 50.57 *	8.81 ± 4.20	1.53 ± 1.36	2.96
G-EO-3	39.68 ± 7.30	3.42 ± 0.63	0.52 ± 1.00	1.15
G-EO-4	38.52 ± 18.30	2.77 ± 1.32	1.75 ± 1.27	0.93
G-EO-5	26.25 ± 22.26	2.70 ± 2.29	2.51 ± 0.94	0.91
G-EO-8	53.54 ± 16.06	4.71 ± 1.41	2.58 ± 0.48	1.58
G-EO-9	61.38 ± 22.90	4.19 ± 1.57	2.40 ± 0.73	1.41
G-EO-19	136.25 ± 21.04 *	11.12 ± 1.72	0.27 ± 1.64	3.73
G-PDM-4	40.86 ± 19.3	3.60 ± 1.70	2.74 ± 0.34	1.38
G-PDM-6	85.21 ± 21.07 *	6.83 ± 1.69	1.46 ± 1.22	2.61
G-PDM-10	87.37 ± 45.77 *	7.76 ± 4.07	2.24 ± 0.67	2.97
G-PDM-19	91.58 ± 18.4 **	5.15 ± 1.03	1.81 ± 1.04	1.97
G-PDM-22	100.24 ± 19.27 **	6.29 ± 1.21	2.40 ± 0.77	2.40
G-PDM -35	51.29 ± 14.04	3.66 ± 1.00	2.38 ± 0.57	1.40
G-PDM 41	36.80 ± 18.45	3.22 ± 1.71	1.90 ± 1.87	1.23

J*_SS_*: steady state flux; Kp: caffeine permeability coefficient; Tlag: lag time; EI: enhancement index = Kp with enhancer/Kp without enhancer; SD: standard deviation * *p* < 0.05, ** *p* < 0.01.

**Table 4 molecules-28-02872-t004:** Plant-derived metabolites (PDM) tested.

Code	Name/Type	Structure	MW * (g/mol)
PDM-4 **	(R)-(+)-Citronellal/Acyclic monoterpenoid	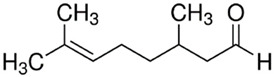	154.25Log P 3.83
PDM-6	(R)-(+)-Limonene/Monocyclicmonoterpene	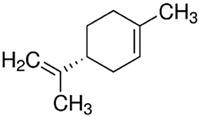	136.23Log P 4.45
PDM-10	1-Bromonaphthalene/Polyaromatic hydrocarbon	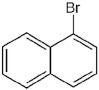	207.07LogP 4.06
PDM-19	Carvacrol/Phenolicmonoterpene	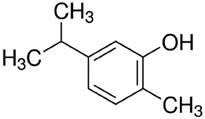	150.22LogP 3.28
PDM-22	Eugenol/Phenilpropanoid	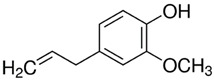	164.20LogP 2.50
PDM-35	α-Phellandrene/Hydrocarbon monoterpene	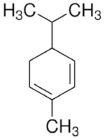	136.23LogP 4.43
PDM-41	trans-β-Caryophyllene/Bicyclic sesquiterpene	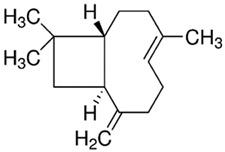	204.35LogP 6.30
Control	Caffeine	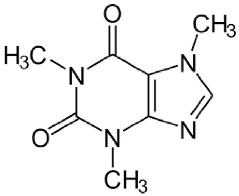	194.19LogP −0.07

* MW: molecular weight; ** PDM: plant-derived metabolite.

## Data Availability

Not applicable.

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
