# Peer review of "Essential Oils Distilled from Colombian Aromatic Plants and Their Constituents as Penetration Enhancers for Transdermal Drug Delivery"

_molecules, 2023, doi:10.3390/molecules28062872_

Round 1

Reviewer 1 Report

This manuscript deals with " Essential oils obtained from Colombian aromatic plants and their constituents as penetration enhancers for transdermal drug delivery". This article claims that using of  essential oils of Colombian aromatic plants could be a suitable for transdermal drug delivery and  treatment agent. The topic is promising and i really enjoyed. Therefore, I suggest a minor correction and require a detailed clarification.

Correction to be addressed by the authors as follows: The abstract is not well organized, where the sentences are incomplete and no continuity is there. It would be feasible, if include the significance of the current study in the abstract. A brief description of how the authors selected information from the literature in the databases, as well as what time period they searched for, is missing. Authors should justify and expand the information on the diseases in which this species is mentioned, highlighting the main contribution in the preclinical and clinical fields.

The major text of this manuscript is focused to some mechanisms but not in the case of pharmacological activities and clinical applications. Authors should specify the main experimental conditions used on the evidences from the literature. Where they briefly describe the most important data reported in the literature in a homogeneous manner and sequence reinforcing the relevance of this species as medicinal alternative. Authors should discuss whether the use of these essential oils is best alternative for currents treatment agents? and also discuss the possibility of using nanoformulation of these agents represents a solid alternative to existing medicines?. Please add more previous studies to your manuscript in discussion section using these papers: -DOI:10.1016/j.arabjc.2021.103106 -DOI:10.1016/j.jcis.2020.10.047   -DOI:10.3389/fbioe.2022.855136

Conclusions should reaffirm the fundamental contribution of this Review.

Author Response

Reviewer 1: The abstract is not well organized, where the sentences are incomplete, and no continuity is there. The significance of the current study It would be feasible, if include in the abstract. A brief description of how the authors selected information from the literature in the databases, as well as what time period they searched for, is missing. Authors should justify and expand the information on the diseases in which this species is mentioned, highlighting the main contribution in the preclinical and clinical fields.

PE: Thanks in advance for all your comments. Some parts of the abstract were completed. Now the abstract followed the order: Aim, material and methods, results, and conclusion. Color blue showed our changes through the manuscript.

Aim: The study aimed to determine the enhancing effect of essential oils (EOs) and plant-derived molecules (PDMs) as penetration enhancers (PEs) for transdermal drug delivery (TDD) of caffeine. Material and methods: A 1% w/w solution of eight EOs and seven PDMs were included in the 1% caffeine carbopol hydrogel. Franz diffusion cell experiments were performed using mice with full-thickness skin. At various times over 24 h, 300 μL of the receptor was withdrawn and replaced with fresh medium. Caffeine was analyzed spectrophotometrically at 272 nm. The skin irritation effects of the hydrogels once a day for 21 days were investigated in mice.

Result: The steady-state flux (JSS) of the caffeine hydrogel was 30±19.6 µg cm-2 h-1. An increase in caffeine JSS was induced by Lippia origanoides > Turnera diffusa > eugenol > carvacrol > limonene with values of 150±14.1, 130±47.6, 101±21.7, 90±18.4, and 86±21.0 µg cm-2 h-1, respectively. The Kp of caffeine was 2.8±0.26 cm h-1 almost 2-4 times lower than induced by L. origanoides > T. diffusa > limonene > eugenol > carvacrol with caffeine Kp values of 11±1.7, 8.8±4.2, 6,8±1.7, 6,3±1,2 and 5,15±1.0 cm h-1 respectively. No irritation signs or skin edema were observed after treatment.

Conclusion: L. origanoides, T. diffusa, eugenol, carvacrol, and limonene improved caffeine skin permeation. These compounds may be effective as the PE of TDD systems.

PE: We did not include how we selected the information from the literature in the databases, and the period because we did not provide a comprehensive review of a specific issue, we discussed in our research paper, our results by topic not by dates. In general, the activities of essential oils obtained from Lippias or other relative aromatic species have been described in some of our works. In this paper, we referred some of our papers as general information. We included only a short phrase about the diseases, especially because the aim was to evaluate those EOs as formulation ingredients able to induce the permeation of actives drugs.  We hope that you agree with us. The references were the followed (the last one was included for your compression):

  1. Stashenko, E.; Martínez, J. R. Study of essential oils obtained from tropical plants grown in Colombia. In Book Essential Oils of Nature, first ed.; Editor El-Shemy, H.A. IntechOpen. 2019, Chapter 8, pp 403-451. doi.org/10.5772/intechopen.87199.
  2. Escobar, P.; Leal, S.M.; Herrera, L.V.; Martinez, J.R.; Stashenko, E. Chemical composition and antiprotozoal activities of Colombian Lippia spp essential oils and their major components. Mem Inst Oswaldo Cruz 2010, 105, 184-190. doi: 10.1590/s0074-02762010000200013.
  3. Stashenko, E.E.; Martínez, J.R.; Ruíz, C.A.; Arias, G.; Durán, C.; Salgar, W.; Cala, M. Lippia origanoides chemotype differentiation based on essential oil GC-MS and principal component analysis. J Sep Sci. 2010, 33, 93-103. doi: 10.1002/jssc.200900452.
  4. Neira, L.F.; Matilla, J.C.; Stashenko. E.E.; Escobar, P. Toxicity, genotoxicity and anti-Leishmania activity of essential oils obtained of four chemotypes of Lippia genus. Bol Latinoam Caribe Plantas Med Aromát. 2018, 17, 68-83. ID: biblio-915131

Reviewer 1: The major text of this manuscript is focused to some mechanisms but not in the case of pharmacological activities and clinical applications.

PE: This is true, this paper was oriented to determine the enhancing effect of essential oils (EOs) and plant-derived molecules (PDMs) as penetration enhancers (PEs) for transdermal drug delivery (TDD) of caffeine. We proposed some EOs and PMD as candidates to be included in topical formulations. The EO or PMDs will be not the active component, however, they will help the entry of actives one. Throughout the article we wanted to show how EOs serve as PE for various drugs which are used in in clinical. The idea of placing in each drug its molecular weight and Log P, was (as stated in the conclusion) the great diversity of drugs (hydrophilic and lipophilic) where natural compounds has the probability serve as PP for TDD.

Reviewer 1: Authors should specify the main experimental conditions used on the evidences from the literature. Where they briefly describe the most important data reported in the literature in a homogeneous manner and sequence reinforcing the relevance of this species as medicinal alternative.

PE: We added in the test: Focus have been centered on identifying effective PE from natural sources. For example, essential oils and terpenes are one of the encouraging groups of candidates to be used as clinically acceptable PE.

One of the most common methodologies for studying transdermal drug delivery in vitro or ex vivo are the diffusion cell systems (Franz cells). They consist of a donor and a receptor compartment, separated by a barrier, i.e., an artificial membrane, or skin sample. Diffusion studies testing EOs or derivatives molecules as skin penetration enhancers for transdermal drug delivery of several drugs have been performed using different type of membranes as excised human, pig, rabbit, rat, mouse, hairless mouse skin, cellulose membrane.

We added the reference:

  1. Franz, T.J. Percutaneous absorption on the relevance of in vitro data. J Invest Dermatol. 1975, 64, 190-195. doi: 10.1111/1523-1747.ep12533356.

The references that you suggest are very interesting for a medicinal plant review, and we will use for sure in other type of paper.

Reviewer 1: Authors should discuss whether the use of these essential oils is best alternative for currents treatment agents?  

In our case, EO will be no alternatives for current treatment agents, however, they could be the best as PE as we comment at the introduction:  

They can enhance drug penetration into the lower skin layers by interacting with lipids and proteins i.e., by disintegration (solubilization, extraction) of intercellular lipid structure between corneocytes in SC, by interaction with an intercellular domain of protein i.e., keratin or by increasing the partitioning of a drug, a coenhancer, water, or any combination of these [5,6,10]. Overall, they could be safe (free of adverse side effects), however, toxicity is dependent on many factors such as their chemical composition, purity (method of extraction and conservation, age), and dosage where a balance of safety versus potency allows their use in the lowest possible concentration [5,6,11].

Reviewer 1: and also discuss the possibility of using nanoformulation of these agents represents a solid alternative to existing medicines?.

PE: We added

Because EO or their constituent are volatile, and easily decomposed by environmental conditions, novel TDD nanosystems such as microemulsion, nanoemulsion, nanoemulgel, liposomes, solid lipid nanoparticles, nanostructured lipid carriers have been designed to increase their effectiveness as skin PE. As an example, both lipophilic and hydrophilic drugs such as caffeine (MW194.2, logP -0.07), naproxen (MW 230.26, logP 3.18), and diclofenac sodium (MW 318.13, LogP 4.75) markedly enhanced skin permeation of using nanoemulsion or nanoemulgel formulations containing eucalyptol, oleic acid, and cumin EO as PE, in comparison to all controls.

17. Abd, E.; Namjoshi, S.; Mohammed, Y.H.; Roberts, M.S.; Grice, J.E. Synergistic skin penetration enhancer and nanoemulsion formulations promote the human epidermal permeation of caffeine and naproxen. J Pharm Sci. 2016, 105, 212-220. doi: 10.1002/jps.24699.

18. Morteza-Semnani, K.; Saeedi, M.; Akbari, J.; Eghbali, M.; Babaei, A.; Hashemi, S.M.H.; Nokhodchi, A. Development of a novel nanoemulgel formulation containing cumin essential oil as skin permeation enhancer. Drug Deliv Transl Res. 2022, 12, 1455-1465. doi: 10.1007/s13346-021-01025-1.

Reviewer 2 Report

Line 237 :  The  codes, source and composition was described in Table 4 not Table 3.

Line119 - 125 : This should be properly spaced

Line 148- 149 and 149- 150 :   a and b should be added to the figure

Line 210: A and B should be put in bracket

Author Response

Line 237 :  The  codes, source and composition was described in Table 4 not Table 3. OK

Line119 - 125 : This should be properly spaced OK

Line 148- 149 and 149- 150 :   a and b should be added to the figure OK

Line 210: A and B should be put in bracket

Author Response

p.10 229-227 Section 4.1 Plant material
The Authors should add a separated paragraph related to the extraction and experimental
analysis specifying what type of plant material (plant part, dry or fresh) and the amount of
plant material used for the extraction.
Table 3
Regarding GC/MS analysis, the Authors did not report the identification parameters such as
LRI or MS target peaks.
In addition, the Authors should provide the GC/MS chromatograms as available
supplementary materials to link the Table 3 ( composition of the tested EO) with their
experimental results and discussion.

OK, all sugestion were performed, please see supplentary material
